# A 2.6 ppm/°C 2.5 V Piece-Wise Compensated Bandgap Reference with Low Beta Bipolar

**Quanwang Liu** [1,2,*]**, Bo Zhang** [1]**, Shaowei Zhen** [1,*]**, Weidong Xue** [2] **and Ming Qiao** [1]

1   State Key Laboratory of Electronic Thin Films and Integrated Devices, University of Electronic Science and Technology of China, Chengdu 610054, China; zhangbo@uestc.edu.cn (B.Z.); qiaoming@uestc.edu.cn (M.Q.)
2   O2micro Co. Ltd., Chengdu 610041, China; xueweidong721@gmail.com
*   Correspondence: cd.liuquanwang@gmail.com (Q.L.); swzhen@uestc.edu.cn (S.Z.);
    Tel.: +86-132-0818-6128 (Q.L.); +86-138-8072-8831 (S.Z.)

**Abstract:** Traditional bandgap reference (BGR) is sensitive to process variation and is not suitable for mass production. Consequently, a stacked piece-wise compensated bandgap reference (SPWBGR) with low beta bipolar is proposed, designed and fabricated in the 0.18 μm high-voltage (HV) BCD process. Two stacked BGR (SBGR) cores make up the proposed BGR circuit. Through setting the target reference voltage near the output voltage of SBGR cores, the feedback resistor ratio is reduced and the base current side-effect is significantly decreased. Notably, the SBGR core is implemented by the low beta npn bipolar and it relaxes the requirement for the high beta bipolar. The two SBGR cores are almost identical except for the temperature slope and feedback ratio. The two cores have different zero temperature coefficient (TC) points, one is set at −5 °C, and the other is set at 60 °C, named as SBGRA and SBGRB, respectively. The SBGRA and SBGRB output the same voltage at their zero TC point. The higher voltage of SBGRA and SBGRB is the output voltage. Through the process of tracking the maximum value of different SBGR cores, the proposed SPWBGR achieves 2.6 ppm/°C TC from −40 to 100 °C. As a result, the average TC for five random samples is 5.3 ppm/°C. The line regulation is 2 mV/V from 4.5 to 5.5 V power supply. The current consumption is 6.8 μA. The active area of the proposed BGR is 0.075 mm².

**Keywords:** piece-wise compensation; bandgap reference; low temperature coefficient; TC

## 1. Introduction

The precision reference voltage is widely used in mixed-signal and analog circuits in a system on chip (SoC). The reference voltage provides accuracy voltage amplitude for battery and power management chips, monitor and supervisory devices, data acquisition systems, etc. A low TC reference is an important integrant of precision data acquisition systems. However, a high accuracy reference cannot get from the first-order compensated BGR directly, since high precision data acquisition systems need the reference with several ppm/°C TC.

The traditional bandgap reference [1,2] can only reach several tens of ppm TC. Many compensation methods are adopted to improve TC recently [3–7]. Two BGR cores summation compensation utilizes the summation of the two balanced curvature-up and curvature-down currents [3]. The second-order compensation utilizes the difference of the two non-linear CTAT voltage [4]. Piece-wise compensation utilizes a piece-wise nonlinear curvature corrected current [5]. Another piece-wise compensation utilizes the exponential compensation in the temperature range and logarithmic compensation in the high temperature range [6]. Another method utilizes two identical first-order bandgaps with different zero temperature points [7]. Other compensation methods, such as VBE linearization [8,9], temperature-dependent resistor ratio compensation [10], resistor-less, successive voltage step

compensation [11], base current compensation [12], subthreshold compensation [13], and so on. However, some of these compensations rely upon the matching of transistor [3,14,15], resistor [3], the high beta bipolar [7], process stability or power supply voltage, etc. Reference [3,5] has no trimming, but its output variation range is large, which is not suitable for high precision measurement. Reference [7] requires high beta npn to obtain a high VREF voltage. For mass production, the multi-point temperature calibration must be implemented [6,8,9,11], therefore the test cost is significantly high.

In order to avoid the multi-point temperature calibration for cost saving, and obtain a precision output voltage, a stacked piece-wise compensated BGR (SPWBGR) is introduced. By setting a small feedback resistor ratio, the offset voltage of the proposed SPWBGR is small, and the side-effect of the base current on TC is ignorable, while the high beta of the npn bipolar is not critical anymore. The SPWBGR utilizes two identical first-order BGR cores, which is easy to implement, and has low sensitivity to process variations.

This is an extension of the piece-Wise compensated BGR posted in the 2017 IEEE 12th International Conference on ASIC (ASICON) [7]. The differences between the new proposed SPWBGR and the previous art noticed are as follows: (1) VREF output voltage amplitude increases to 2560 mV from 1536 mV, as the reference voltage for Successive Approximation ADC. (2) SPWBGR reduces the requirement for high beta npn bipolar by decreasing the feedback resistor ratio, the previous art uses high beta npn to relieve the side effect of the base current. (3) The offset voltage of SPWBGR is reduced through a small feedback resistor ratio. In the previous art, the offset voltage is amplified by a large feedback resistor ratio. (4) Current consumption of the two stages of SBGR decreases by removing the bias branch that generates the cascode bias. (5) The layout area is significantly reduced by optimizing the device size.

## 2. Background

In the battery protection and monitor system depicted in Figure 1 the status of battery cells is monitored, which includes aspects such as voltage, current, temperature, etc. These parameters are sensed and converted to a voltage signal, which is then sampled and converted by the SAR ADC. The bandgap reference VREF is the reference voltage for the 14-bit SAR ADC. Notably, the TC of VREF will directly affect the accuracy of ADC. In order to sense the battery cell voltage, the ratio of the maximum battery cell voltage to the amplitude of VREF, determines the scale-down ratio. For example, the Li-ion/Polymer battery cell voltage is up to 5 V, which is sensed by the SAR ADC with the scale-down ratio. The ratio is two for VREF = 2.56 V and the ratio is three for VREF = 1.536 V. Notably, the reference voltage determines the full-scale range of the SAR ADC. The scale-down ratio is inverse proportional to the reference voltage. Moreover, the large scale-down ratio will introduce an extra error for the ADC conversion, this is especially because the offset voltage is amplified by the scale-down factor. Consequently, the SPWBGR is more suitable for this application.

The base current of npn bipolar flows through the resistors, and it will introduce an extra voltage drop while also affecting the TC. High beta npn is used to decrease the base current IR drop in the previous art and its beta is up to 100. In this HV BCD process, the beta of npn is only expressed more than a dozen. If the structure in the previous art is used, which is shown in Figure 2. The VREF voltage is 2.56 V higher than the previous 1.536 V. The base current causes an extra IR drop on R4, which will affect the TC of VREF to be much higher than the previous art. Even worse, the offset voltage at the base of QN2 is amplified by the ratio (R4 + R5)/R5.

In order to solve the side effect of the base current, a new two-stages SBGR is proposed. The output of SBGR is approximately 2.4 V, which is near the 2.56 V target output. The resistor ratio R4/R5 is approximately 0.067. The base current flows through the small R4, so the offset voltage of SBGR is only amplified by a small ratio.

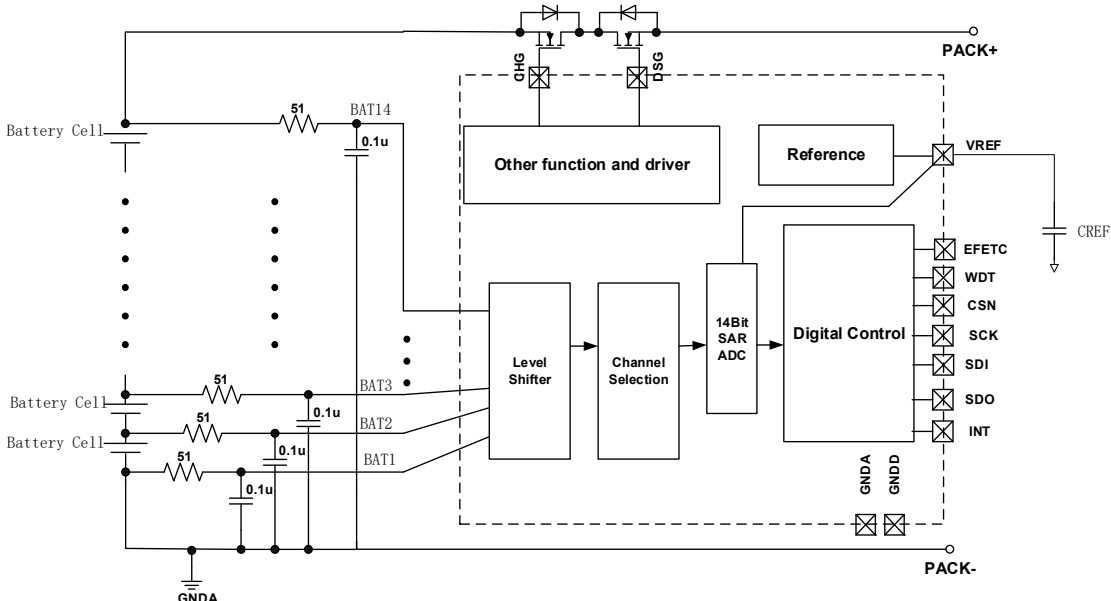

**Figure 1.** Application diagram of battery protection and monitor system.

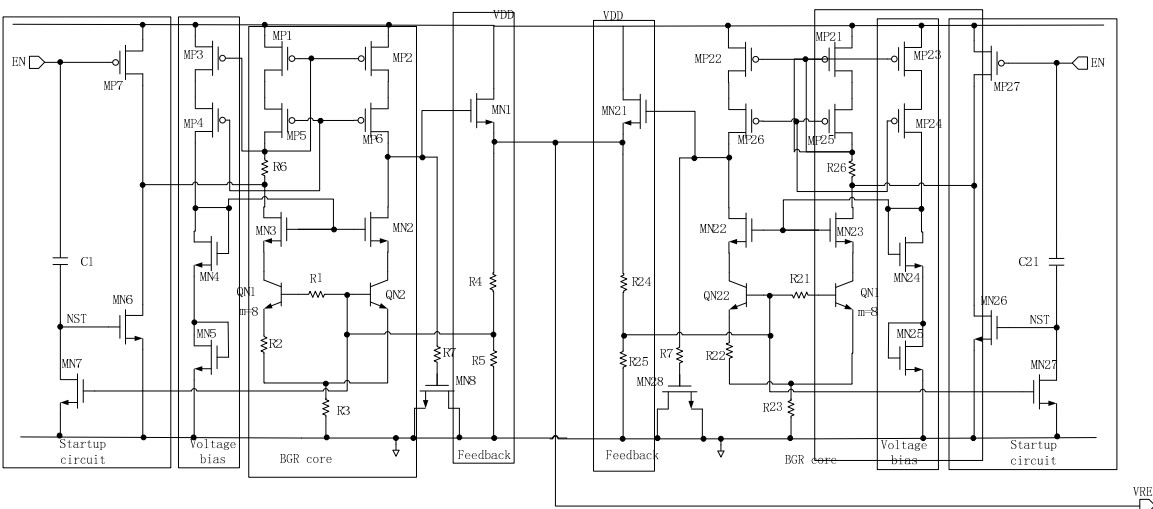

**Figure 2.** Previous art [7].

## 3. Design of SPWBGR

Figure 3 reflects the concept of the proposed SPWBGR. Two identical SBGR cores (SBGRA and SBGRB) make up the proposed SPWBGR and the output of them connect together. The two SBGR cores are only the first-order compensated BGR, which has an output voltage of approximately 2.4 V. The zero-temperature point for SBGRA is set to –5 °C, while SBGRB is set to 65 °C.

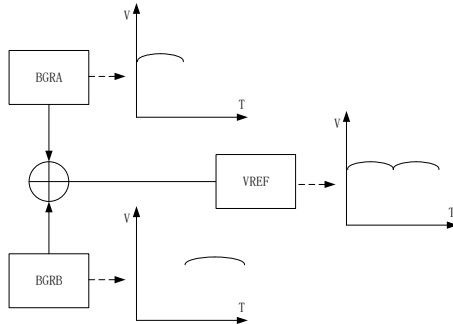

**Figure 3.** Concept of proposed stacked piece-wise compensated bandgap reference (SPWBGR).

Figure 4 shows the diagram of the proposed two-stages SPWBGR core. Assuming that SPWBGR only has SBGRA core on and the base current of QN1 to QN4 are neglected, the output voltage is as follows:

$$V_{REF}(T) = V_{V2BGA}(T)(1 + \frac{R_4}{R_5}) \tag{1}$$

Assuming $I_{MPB1} = I_{MPB2}$ and $Q_{N3}/Q_{N2} = Q_{N2}/Q_{N1}$ ratio is 5, it can get the voltage of $V_{R3}$

$$V_{R3}(T) = \frac{V_{BE3}(T) + V_{BE2}(T) - V_{BE4}(T) - V_{BE1}(T)}{R_{1A} + R_{1B}} \cdot 2R_3 \tag{2}$$

$$V_{BEx(T)} = V_{G0} + [V_{BEx}(T_0) - V_{G0}]\frac{T}{T_0} + \frac{kT}{q}[\ln\frac{I_{CQx}(T)}{I_{CQx}(T_0)} - \eta\ln\frac{T}{T_0}] \tag{3}$$

where k = 1.38 × 10$^{-23}$ J/K, it is Boltzmann's constant, x corresponds to 1 and 2, and $V_{G0}$ is 0 K bandgap voltage of silicon, η is a constant related to technology, $T_0$ is the temperature reference, T is the absolute temperature

$$\Delta V_{BE}(T) = V_{BE3}(T) + V_{BE2}(T) - V_{BE4}(T) - V_{BE1}(T) = 2\frac{kT}{q}\ln N \tag{4}$$

where N is the ratio of the emitter area, and in the design, set N = 5.

Taking Equation (1) derivative with respect to temperature, the ratio of $R_3/ (R_{1A} + R_{1B})$ is obtained at $T_0 = -5\,°C$

$$\frac{dV_{VREF}(T)}{dT}|T_0 = (\frac{V_{BEx}(T_0) - V_{G0}}{T_0} + \frac{R_3}{R_{1A} + R_{1B}} \cdot 2\frac{k}{q}\ln N) \cdot (1 + \frac{R_4}{R_5}) = 0 \tag{5}$$

From Equation (5), we obtain

$$\frac{R_3}{R_{1A} + R_{1B}} = \frac{q}{kT_0}\frac{V_{BEx}(T_0) - V_{G0}}{2\ln N} \tag{6}$$

In contrast, assuming SPWBGR only has SBGRB core on. SBGRB is optimized at $T_0 = 65\,°C$, it can obtain a similar ratio.

When SBGRA and SBGRB are both on, MN1 and MN2 have different gate voltage. Therefore, the higher gate voltage determines the SPWBGR output voltage.

Consequently, the proposed SPWBGR output voltage is as follows:

$$V_{REF}(T) = Max(V_{REFA}(T), V_{REFB}(T)) \tag{7}$$

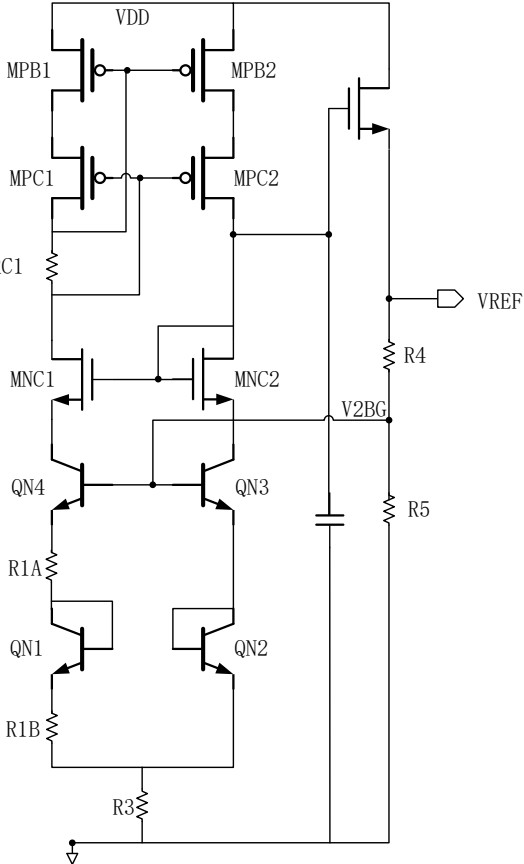

**Figure 4.** Proposed bandgap core diagram.

In the –40 °C ~30 °C temperature range, SBGRA determines the VREF output; in 30 °C ~100 °C temperature range, SBGRB determines the VREF output.

The proposed SBGR intellectually selects the higher SBGR core as the output voltage. In –40 °C to 30 °C temperature range, SBGR selects the SBGRA output; in 30 °C to 100 °C temperature range, SBGR selects the SBGRB output.

Figure 5 indicates the SBGR core detailed schematic. MPB1, MPC1 and RC1, compose the current mirror with wide swing. MNC1 and MNC2 combine as a source follower, so that the collector voltages of QN3 and QN4 are the same. R1A and R1B determine the proportional To absolute temperature (PTAT) magnitude of the current. Such stacked structure can generate a two times bandgap voltage reference, which is approximately 2.4 V, and is much closer to the target output voltage of 2.56 V. The startup circuit is realized by RST, MNST1 and MNST2, and VSTOFF is a proper voltage divided by the voltage of R5.

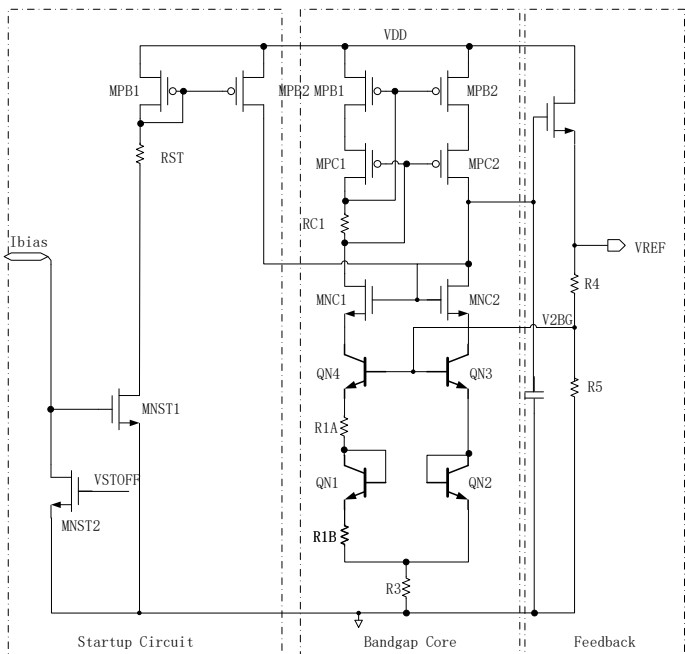

**Figure 5.** Detailed stacked bandgap reference (SBGR) core circuit.

Consider the base current $I_B$ affection on the reference voltage, assuming offset voltage at node V2BG is $V_{OS}$, Equation (1) changes to Equation (8):

$$V_{REFIB}(T) = V_{REFIB}(T) + I_B(T)R_4(T) \tag{8}$$

where $I_B(T)$ is the sum of base current of QN3 and QN4, assuming β of QN3 and QN4 are the same, we can obtain:

$$I_{B2}(T) = \frac{V_{BE\_QN3} + V_{BE\_QN2} - V_{BE\_QN4} + V_{BE\_QN1}}{\beta_2(R_{1A} + R_{1B})} \times 2 \tag{9}$$

Taking (8) derivative with respect to the temperature, it can yield Equation (10)

$$\frac{dV_{VREFIB}(T)}{dT}\Big|T_0 = \frac{dV_{VREF}(T)}{dT}\Big|T_0 + I_B(T_0)\frac{dR_4(T)}{dT} + R_4(T_0)\frac{dI_B(T)}{dT} \tag{10}$$

From Equation (10), the extra temperature coefficient comes from $I_B$ and $R_4$. $I_B$ is inverse proportional to the beta of npn bipolar, and $I_B$ is very small in several tens of nA in this work. R4 is small for the two-stages SBGR, therefore the extra effect introduced by the base current and R4 can be ignored.

For the one stage structure in the previous art 7, in a similar way, it can obtain a similar formula, but R4 in single BGR is dozens of times larger than R4 in SBGR. This is especially since its effect on TC cannot be ignored anymore.

## 4. Implementation of the Proposed SPWBGR

The proposed SPWBGR is implemented in a 0.18 μm high-voltage (HV) BCD process with 5 V devices. The proposed SPWBGR intelligently selects the output of two SBGR cores. SBGRA and SBGRB have slight differences with the same structure, the difference is R3, R23, R4 and R24 as shown in Figure 6. Due to the fact that SBGRA and SBGRB zero TC point is set at –5 °C and 65 °C respectively, R3/(R1B + R1A) < R23/(R21B + R21A) with a small difference. SBGRA and SBGRB output the same voltage at zero TC point when R3/(R1A + R1B) and R23/(R21A + R21B) are different. Thus, R4/R5 > R24/R25 with a small difference.

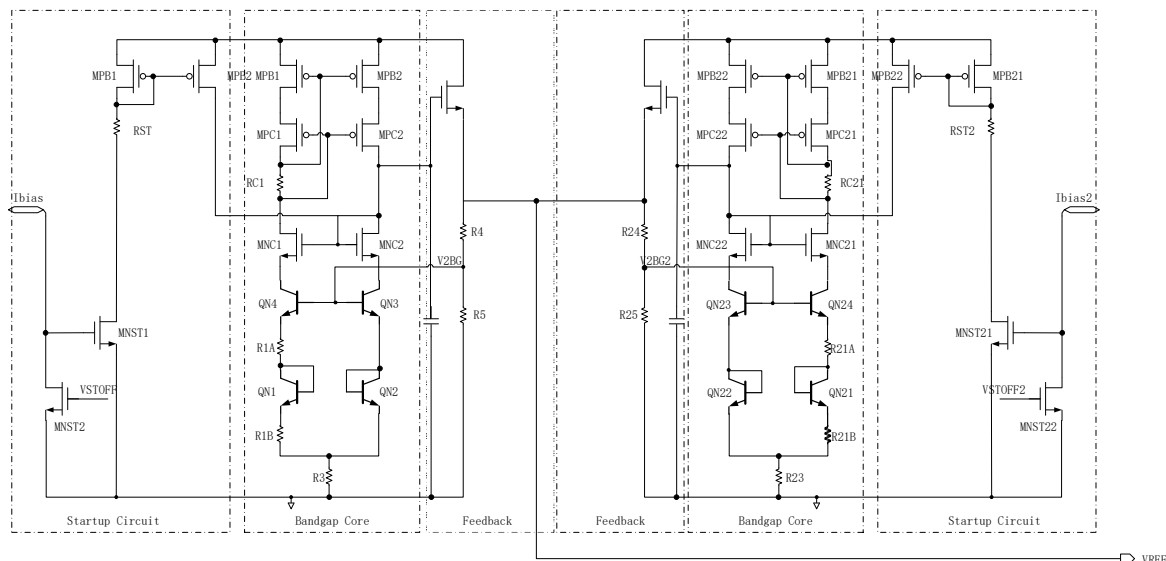

**Figure 6.** Proposed SPWBGR schematic.

When VREF is fixed, from Equations (1) and (2), we obtain that the ratio of R4/R5 and that of R23/(R1B + R1A) are an inverse proportional relationship. In a similar way, R24/R25 is inverse proportional to R23/(R21B + R21A).

Table 1 presents SPWBGR device sizes in Figure 6. SBGRA and SBGRB have the same sizes, except R3, R23, R4 and R24. Consequently, the matching of SBGRA and SBGRB are good. All resistors are generated by a series connection as well as a parallel connection of the unit resistor.

**Table 1.** The proposed SPWBGR device size table.

| Components | Parameter (m) |
|---|---|
| MPB1, MPB2 MP21, MP22 | W = 2 μ, L = 20 μ |
| QN2, QN3, QN22, QN23 | m = 1 |
| MPC1, MPC2, MPC21, MPC22 | W = 10 μ, L = 2 μ |
| MNC1, MNC2, MNC21, MNC22 | W = 10 μ, L = 1 μ |
| MNST1, MNST2 | W = 1 μ, L = 5 μ |
| QN1, QN4, QN21, QN24 | m =5 |
| R1A, R21A | number of series:1.5 Unit: W = 1.5 μ, L = 25 μ |
| R1B, R21B | Unit: W = 1.5 μ, L = 25 μ number of series: 2.5 |
| R3 | Unit: W = 1.5 μ, L = 25 μ number of series:61 |
| R4 | Unit: W = 1.5 μ, L = 6.25 μ |
| R23 | Unit: W = 1.5 μ, L = 25 μ number of series:63.5 |
| R24 | number of series:8.125 Unit: W = 1.5 μ, L = 6.25 μ |
| R5, R25 | Unit: W = 1.5 μ, L = 35 μ number of series: 23 |

## 5. Simulated Results

Figure 7 shows SPWBGR simulation results of different corners. In the typical corner, VREF has 1.73 mV peak to peak difference. The two SBGR cores have the same output voltage at their zero TC point.

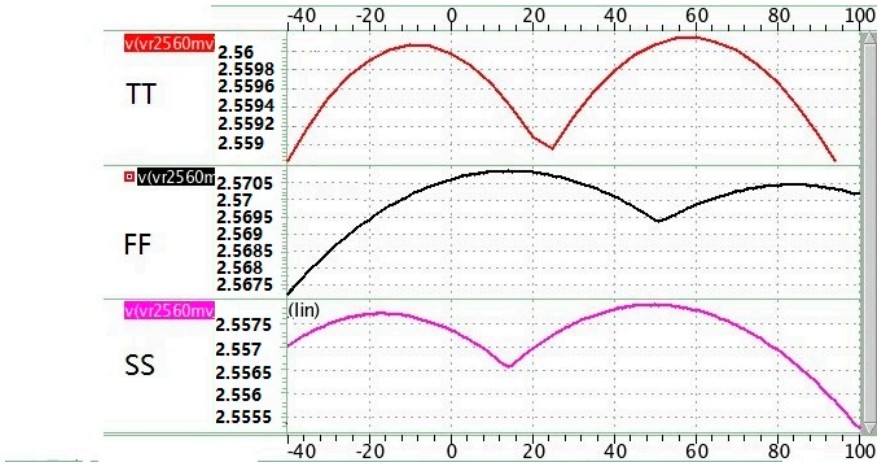

**Figure 7.** Simulation results of proposed SPWBGR.

Figure 8 shows the micrograph of the proposed SPWBGR. SBGRA and SBGRB which are well matched. Figure 9 reflects the layout view of SPWBGR which shows that the layout area is approximately $312 \times 239 \ \mu m^2$, which is much less than the previous art $723 \times 576 \ \mu m^2$, this is only 18% of the previous art. Figure 10 shows the Monte Carlo analysis of the proposed SPWBGR, from the simulation results, the amplitude of the two SBGR cores will variate at different temperatures and it can be corrected by voltage amplitude trimming.

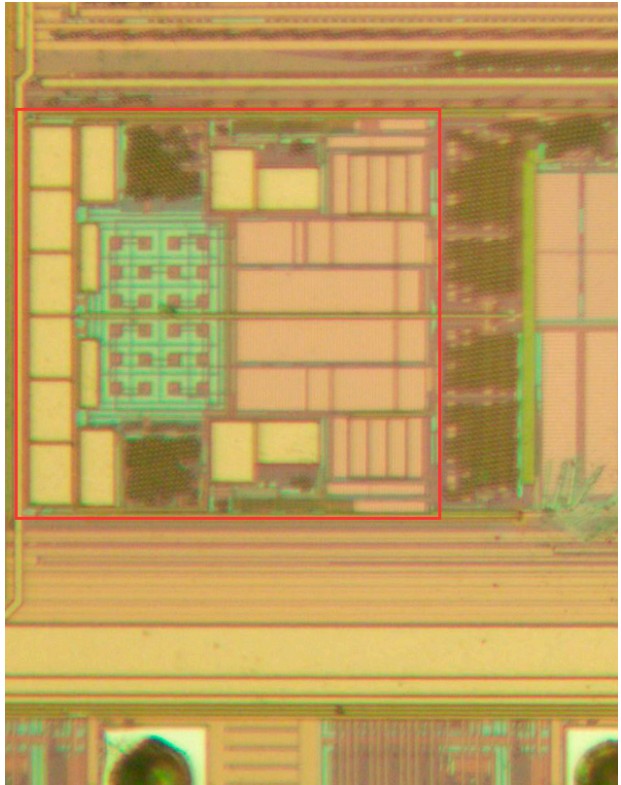

**Figure 8.** Micrograph of proposed SPWBGR.

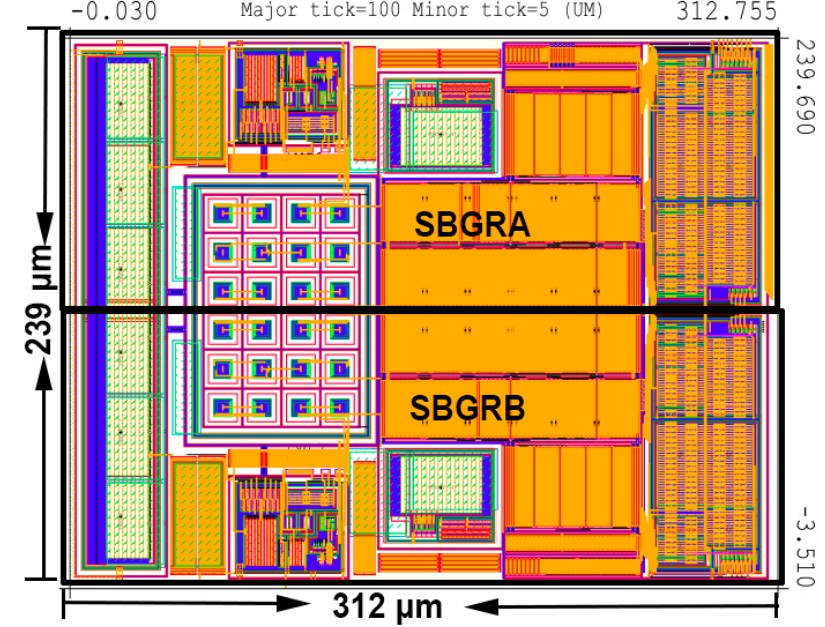

**Figure 9.** Layout of proposed SPWBGR.

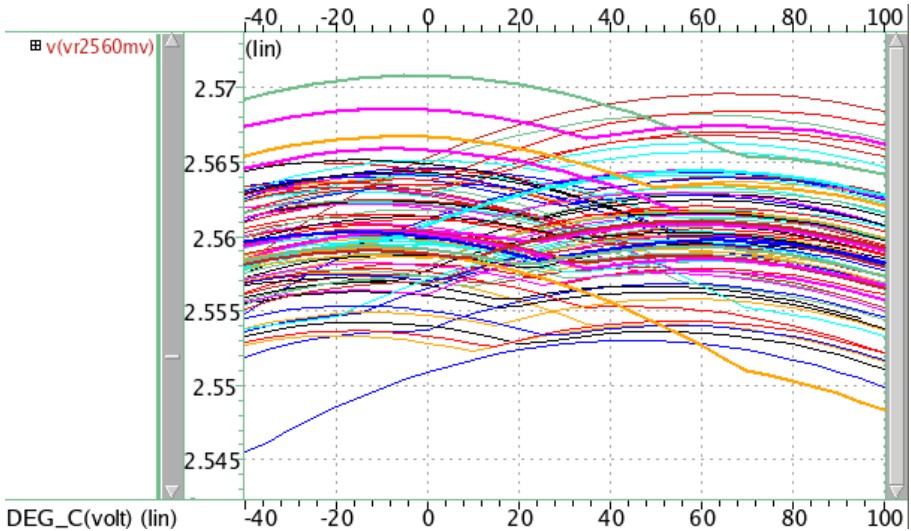

**Figure 10.** Monte Carlo simulation results of proposed SPWBGR.

In Figure 11, the Power Supply Rejection Ratio (PSRR) simulation results of SPWBGR are shown at different corners and temperatures. The proposed BGR has a –63 dB PSRR at 10 Hz with 5 V power supply.

Figure 12 depicts the line regulation simulation results at different corners when VDDA is 4.5–5.5 V. The worst line regulation is 2 mV/V at FF corner, temperature = –40 °C.

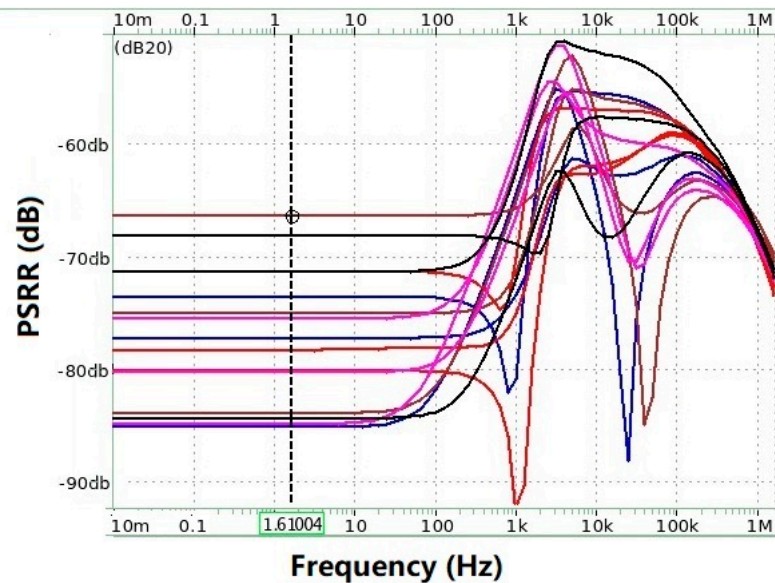

**Figure 11.** Power supply rejection ratio (PSRR) simulation results of proposed SPWBGR.

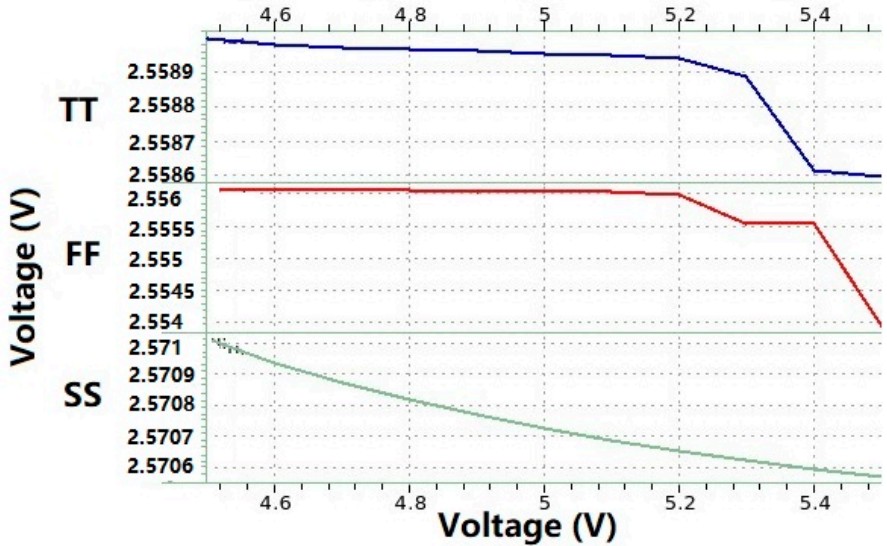

**Figure 12.** Line Regulation simulation results of the proposed bandgap reference (BGR).

## 6. Measurement Results

Figure 13 shows the measured TC of VREF after trimming at room temperature. When both of SBGRA and SBGRB are on, the variation range is 0.9 mV from –40~100 °C. Simply put, it is 2.6 ppm/°C. The SBGRA and SBGRB output voltage cross at 30 °C. In –40 °C~30 °C temperature range, SBGRA output is higher than SBGRB's; in 30 °C~100 °C temperature range, the SBGRB output is higher than the SBGRA output.

Figure 14 reflects five random samples after room temperature trimming. All samples show stable TC between 2558.8 mV to 2560.3 mV after room temperature trimming. TC is 4.2 ppm at average level, and it is considered as an efficient level.

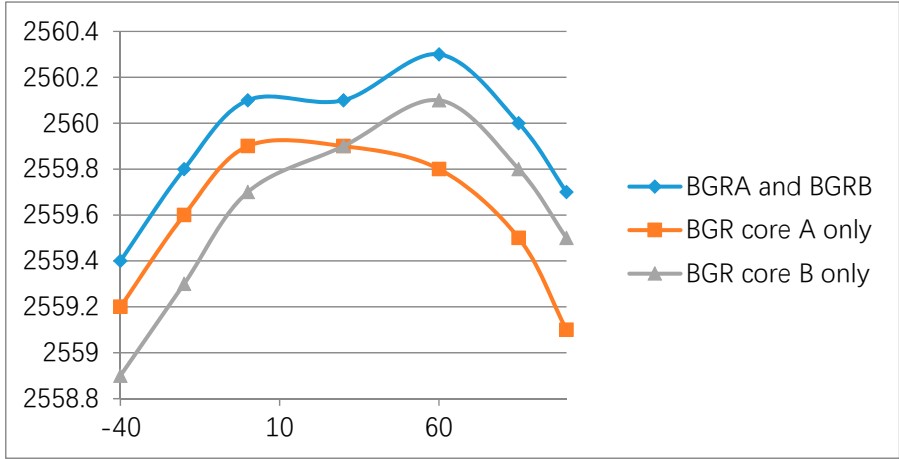

**Figure 13.** Bandgap reference test results.

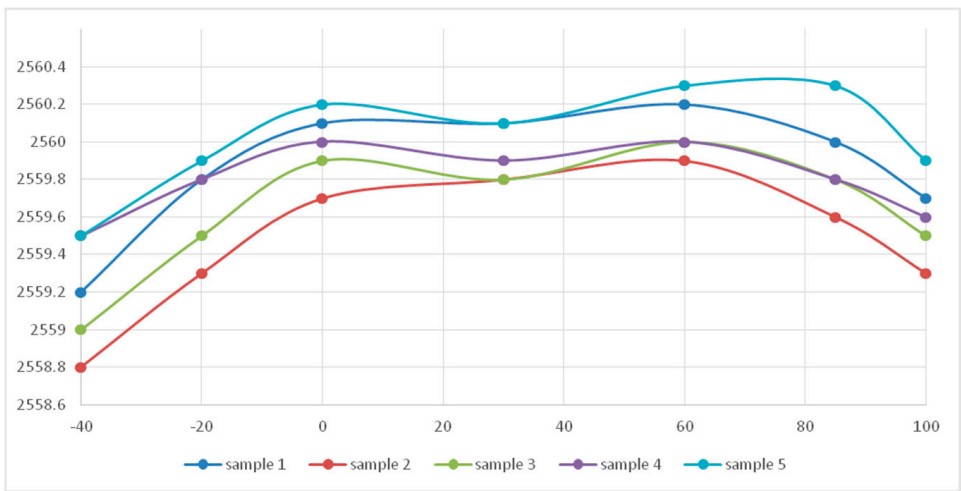

**Figure 14.** Five random samples measurement results.

Table 2 reflects the comparison between this work with other previous work [3,5,6,9–11]. Reference [9] presents superior TC, but requires three-point temperature trims which add a significant test cost. A ppm metric normalizes TC with respect to the VREF voltage. For the battery protection and monitor system, lower VREF voltage must be scaled down further with respect to the battery voltages, so that absolute voltage errors are increased by the scale factor, but TC remains unchanged. Reference [3,5] has no trimming, but the variation range of VREF is large, which is not suitable for the absolute voltage measurement. Reference [10] has good TC with a single-trim, but its current consumption is large. This work achieves the minimum current consumption.

**Table 2.** Performance summary and comparison with other voltage reference.

| Type | Proposed | Q. Duan et al [3] | J. H et al [5] | Z.Zhou et al [6] | Z.Zhou et al [9] | B. L. [10] | X. Ming et al [11] |
|---|---|---|---|---|---|---|---|
| Technology | 0.18 μm HV CMOS | 0.13-μm CMOS | 0.5-μm CMOS | 0.5-μm BiCMOS | 0.35-μm CMOS | 0.18-μm BiCMOS | 0.5-μm CMOS |
| Temperature range (°C) | −40~100 | −40~120 | −40~110 | −40~100 | −40~125 | −40~110 | −10~130 |
| Trimming method | Single trim | No-trim | No-trim | Multi-trim | Multi-trim | Single trim | Multi-trim |
| Supply Voltage | 4.5–5.5 | 1.2 | 5 | 1.6–5 | 2–5 | 5.2 | 2.1–5 |
| VREF (V) | 2.56 | 735 | 1.2 | 1.285 | 1.14055 | 3.65 | 1.196 |
| Current consumption (μA) | 6.8 | 120 | 9.6 | 25 | 33 | 750 | 38 |
| Min. TC (ppm/°C) | 2.6 | 4.2 | 8.9 | 5 | 1.01 | 3 | 3.98 |
| Line regulation (mV/V) | 2 [1] | N.A. | 2.4 | 0.35 | 2 | N.A. | 0.19 |
| PSRR | −63[S] | −30 | −58 | −70 | −61 | −127 | −84 |
| Chip Area (mm$^2$) | 0.075 | 0.063 | 0.1 | 0.04 | 0.0396 | 0.28 | 0.053 |

[1] Simulation results.

## 7. Conclusions

A two-stage SPWBGR is introduced in the paper. The SPWBGR reduces the side effect of the base current and offset voltage by setting a small feedback resistor ratio. Through room temperature trimming, the test cost is significantly reduced. The current consumption is 6.8 μA at the typical corner. It can get lower TC by more SBGR cores.

**Author Contributions:** Q.L. developed the math model and managed the preparation of the manuscript; B.Z. provided the supervision; S.Z. participated in the math equations and solutions; W.X. provided the idea of the research and collaborated in math solution verification; M.Q. provided the investigation of the research

**Funding:** This research has not received external funding.

**Conflicts of Interest:** The authors declare no conflict of interest.

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
