# Peer review of "A 2.6 ppm/°C 2.5 V Piece-Wise Compensated Bandgap Reference with Low Beta Bipolar"

_electronics, doi:10.3390/electronics8050555_

Round 1

Reviewer 1 Report

A Stacked BGR designed and fabricated in 0.18μm process, in order to decrease the base current with a low β npn BJT. Sample have a stable reference voltage with the variations of temperature. In my point of view the proposed technique is of high significance, the presentation is of high quality and the paper could be publised in its present form.

Author Response

Response 1: The text has been reviewed again; some grammar and syntax errors are corrected.

Reviewer 2 Report

The paper extends a BGR topology previously reported by the same authors in [7]. Additional material is sufficient for publication. Find in the following this reviewer’s main concern: 1) A more detailed comparison of the proposed technique with [3]-[7] is suggested in the Introduction. In particular, what the authors state regarding multi-point calibration is not true. E.g., solution reported in [5] does not require trimming. 2) Many important data are missing: technology adopted, power consumption, PSR, line regulation 3) Comparison with the state of the art should be added (e.g., adding a table that summarizes experimental results and main characteristics of the various solutions). 4) Revise English throughout the paper

Author Response

Dear Reviewer,

     Thanks for your review and concerns. 

     I have updated them in the new manuscript, and please find my response as attached.

Reviewer 3 Report

This work presents a new design of BGR for the use of high-quality references. The manuscript is logically presented and the methodologies from theoretical to experimental results were appropriate. The method of measuring is also good. Reviewer found this study work interesting and valuable. There is no technical comment to the authors, escept of the request for  a proofreading your submitted manuscript.    

Author Response

(The authors gave the same response as above.)

Round 2

Reviewer 2 Report

The authors have positevely answered to the reviewer's concern

This manuscript is a resubmission of an earlier submission. The following is a list of the peer review reports and author responses from that submission.